# ANIMATING THE STILL: PHYSICS-BASED 3D CINEMAGRAPH FROM MULTI-VIEW IMAGES

## ABSTRACT

3D Cinemagraphs aim to generate visually compelling media by introducing subtle and continuous motion into otherwise static images. Recent efforts have explored this task through 3D reconstruction techniques, but they often fall short in delivering physically plausible and controllable animations. In this paper, we propose a novel physics-driven framework built upon 3D Gaussian Splatting (3D-GS) to address these limitations. Given multi-view images and a user-specified force, our approach first reconstructs a 3D scene using 3D-GS, then embeds the reconstruction into a physically consistent simulation environment. By modeling external and internal force fields and performing accurate force analysis within the reconstructed 3D space, we synthesize fine-grained and interpretable motion that aligns with physical intuition. Our method allows users to intuitively control motion effects via high-level physical parameters, achieving a delicate balance between realism and artistic flexibility. Extensive experiments under diverse force conditions demonstrate that our approach produces stable, interpretable, and visually appealing results, surpassing prior methods in both robustness and controllability.

## 1 INTRODUCTION

Cinemagraphs are a form of hybrid visual media that combine the static appearance of photographs with subtle, seamlessly looping motion in specific regions, creating a striking visual effect widely utilized in art, advertising, and social media to capture attention. Traditionally, cinemagraphs are manually crafted from video sequences by selectively freezing parts of the frame while allowing specific regions to animate.

Recent research has explored automatic cinemagraph generation from a single image. Some approaches synthesize motion by learning pixel-wise displacements through deep generative networks, while others leverage estimated depth maps and optical flows to animate foreground regions. However, existing methods often rely on learned priors that may not generalize across diverse scenes or fail to model motion causality explicitly, leading to animations lack interpretability or controllability. The absence of physical constraints can result in motions that violate fundamental physical principles, such as spatial inconsistency, unrealistic deformations, or implausible force propagation, leading to artificial and less controllable animations.

To address these limitations, we propose a novel physics-driven framework for 3D cinemagraph generation from multi-view images. Our method first utilizes 3D Gaussian Splatting (3D-GS) to reconstruct a view-consistent representation of the scene, then a physically consistent simulation environment is integrated, which supports user-defined external force fields. In addition, we also incorporate internal force modeling inspired by structural mechanics—specifically, elastic and damping forces—to regulate motion propagation and maintain coherence. Each 3D Gaussian is treated as a mass point, and its acceleration is computed by aggregating external and internal forces, enabling physically plausible motion estimation. To improve computational efficiency and temporal smoothness, we further introduce a force propagation mechanism that accelerates motion field generation through localized force transmission across the Gaussian graph. Consequently, our framework produces spatially coherent, visually compelling, and highly controllable motion effects that can be interpretable through classical mechanics. This design also supports interactive control, enabling

users to customize motion behaviors by adjusting intuitive physical parameters such as force direction, magnitude, and area of influence.

To summarize, our key contributions are as follows:

- We incorporate physically inspired structural stress modeling into the cinemagraph generation via 3D Gaussian Splatting. This allows us to simulate physically meaningful dynamics in a way that is both interpretable and controllable, laying the foundation for physically grounded motion generation in 3D Gaussian scenes.
- We orchestrate a unified and efficient pipeline that integrates physical simulation with 3D reconstruction, enabling diverse user-defined force fields—such as spiral, wind fields—to drive expressive and customizable motion synthesis from a single image.
- We demonstrate that our method produces visually compelling, spatially coherent, and physically interpretable cinemagraphs under a wide range of force conditions and scene types, showcasing its robustness and generalizability.

## 2 RELATED WORKS

### 2.1 CINEMAGRAPH

Cinemagraphs achieve a compelling visual effect by keeping most of a scene static while allowing a localized region to loop continuously. This hybrid format has gained popularity in art, advertising, and storytelling due to its subtle motion and aesthetic appeal. While digital tools enable photorealistic cinemagraph creation, the process remains labor-intensive and requires significant manual input, prompting active research into automation. Early methods relied on video inputs to identify cyclic motion regions for seamless looping Guo et al. (2024); Tompkin et al. (2011); Yeh & Li (2012); Liao et al. (2015). Subsequent approaches explored single-image-based generation. For example, (Chuang et al., 2005) proposed a semi-automated pipeline using layer-based segmentation and stochastic motion textures, while (Lin et al., 2018) animated specific elements like waterfalls. More recently, generative models such as StyleGAN Karras et al. (2019) and Text2Cinemagraph Mahapatra et al. (2023) have been employed to synthesize cinemagraphs from images or textual descriptions. To enhance realism, several works extend cinemagraph generation to the 3D domain. For instance, (Li et al., 2023) introduces parallax-rich animations via dense depth maps and scene flow.

### 2.2 3D GAUSSIAN SPLATTING

3D Gaussian Splatting (3D-GS) Kerbl et al. (2023) has emerged as a breakthrough neural representation technique, offering superior rendering quality and efficiency. 3D-GS models the scene using a set of semi-transparent, anisotropic Gaussian ellipsoids. This explicit formulation enables a differentiable rasterization-based pipeline, allowing real-time, high-fidelity rendering while improving geometric accuracy and efficiency. Research directions include improving reconstruction quality—by reducing artifacts, enhancing geometry, or refining rendering techniques Condor et al. (2025); Jiang et al. (2023); Liu et al. (2024b); Kerbl et al. (2024); Zhou et al. (2024b)—as well as accelerating training and inference Lu et al. (2024); Cong et al. (2025); Zhou et al. (2024a). Furthermore, 3D-GS has been widely adopted in generative applications, particularly in AIGC, for producing realistic images, videos, and novel views Lin et al. (2025a); Yang & Wang (2024); Yi et al. (2024); Guizilini et al. (2025); Yang et al. (2024).

### 2.3 PHYSICAL SIMULATION

Physical simulation in graphics has produced many foundational techniques. The mass-spring system models objects as discrete mass points connected by elastic forces; it is simple and efficient but prone to numerical instability. The finite element method (FEM) also discretizes objects but models their Lagrangian properties with approximated basis functions, converting continuous systems into solvable finite representations. While FEM can handle complex structures, its computational cost grows rapidly with simulation complexity. The Material Point Method (MPM) Sulsky et al. (1995) combines Lagrangian particles and Eulerian grids: particles transfer physical states to a mesh, equations are solved on the grid, and updates are mapped back to the particles. The emergence of 3D

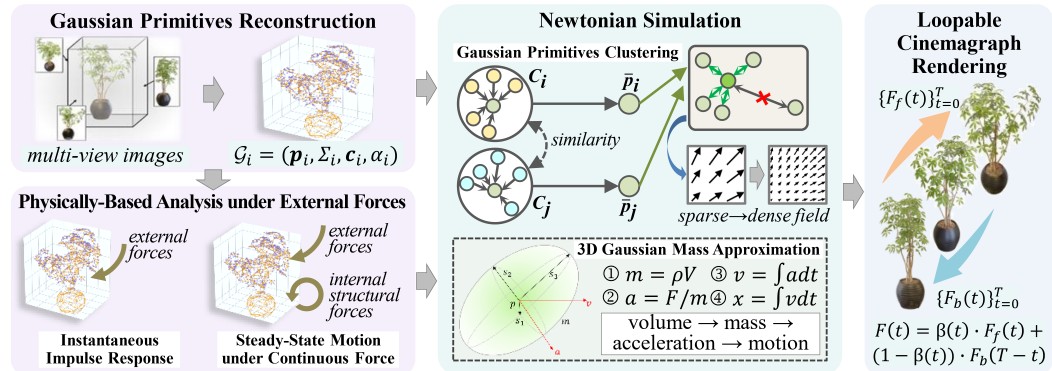

Figure 1: **Framework Overview**. Given multi-view images, we reconstruct a 3D scene using anisotropic Gaussian primitives. These Gaussians are then grouped into SuperGaussians for physically-based simulation under both external and internal forces. To improve structural coherence, we elevate the simulation granularity from individual Gaussians to SuperGaussians—formed via clustering in spatial-feature space. Internal structural forces are modeled through a constraint graph that connects only relevant neighbors. Finally, we employ a loopable rendering strategy that blends forward and backward trajectories, generating seamless 3D cinemagraphs with realistic and temporally coherent motion.

Gaussian Splatting (3DGS) has sparked new directions for physics-integrated animation. PhysGaussian Xie et al. (2023) first applied MPM to 3DGS, but lacked shadow rendering and required manual tuning. VR-GSJiang et al. (2024) adds shadow mapping but retains fixed material presets. Recent work automates material learning: PhysDreamer Zhang et al. (2024) uses MLPs and diffusion models to infer stiffness from text or image prompts; Physics3D Liu et al. (2024a) optimizes MPM pipelines. Other methods go beyond MPM: GASP Borycki et al. (2024) encodes Gaussians into triangle-based structures for simulation, Spring-GauS Zhong et al. (2024) introduces explicit force systems, and DreamGaussian4D Ren et al. (2023) models motion using MLPs with 4D hexplane features. OmniPhysGS Lin et al. (2025b) introduces continuum mechanics to Gaussian tasks, it predicts the motion using a neural network under physical prior. However, challenges remain in supporting real-time shadows, adaptively modeling material diversity, and scaling to complex scenes.

## 3 METHOD

Our full pipeline is illustrated in Fig. 1. We begin by reconstructing the volumetric representation of the scene using 3D Gaussian primitives, and then simulate motion through a Newtonian-based framework tailored for Gaussian-bounded systems, producing time-varying trajectories for each primitive. Finally, we apply a loopable rendering strategy to synthesize temporally seamless frames, which are then concatenated into a continuous animation. To enhance simulation quality and structural coherence, we introduce two key optimizations: (1) We group individual gaussians into coherent clusters using the SuperGaussian method, enabling more stable and interpretable motion behaviors; (2) We define a bone-like structural constraint graph that specifies which Gaussian pairs exert internal forces, improving deformation control and physical plausibility.

### 3.1 GAUSSIAN PRIMITIVES RECONSTRUCTION

Given a set of multi-view images $\{I_k\}_{k=1}^K$, we aim to reconstruct the underlying 3D scene using a collection of *anisotropic 3D Gaussian primitives*. Each primitive is defined by both spatial and radiometric attributes, and the full set is denoted as:

$$\mathcal{G} = \{\mathcal{G}_i\}_{i=1}^N, \quad \text{where} \quad \mathcal{G}_i = (\boldsymbol{p}_i, \Sigma_i, \boldsymbol{c}_i, \alpha_i), \tag{1}$$

with $\boldsymbol{p}_i \in \mathbb{R}^3$ representing the 3D center, $\Sigma_i \in \mathbb{R}^{3 \times 3}$ a positive-definite covariance matrix encoding anisotropic shape and orientation, $\boldsymbol{c}_i \in \mathbb{R}^3$ the RGB color vector, and $\alpha_i \in [0, 1]$ the opacity

coefficient. Each image $I_k$ is assumed to be generated by projecting the 3D Gaussian primitives onto a 2D image plane using a view-dependent projection function $\Pi$, which incorporates camera pose and perspective transformation.

To render each image $I_k$, we apply front-to-back alpha compositing over the projected 2D Gaussians. The color at pixel $(u, v)$ is computed as similar as rendering method of NeRF Mildenhall et al. (2021). The opacity at pixel $(u, v)$: $T_i(u, v)$ is the accumulated transmittance up to (but not including) the $i$-th Gaussian:

$$T_i(u, v) = \prod_{j < i} \left( 1 - \alpha_j \cdot \mathcal{N}\left( (u, v); \boldsymbol{\mu}_j^{\text{img}}, \Sigma_j^{\text{img}} \right) \right). \tag{2}$$

Where $\mathcal{N}((u, v); \boldsymbol{p}, \Sigma)$ is a 2D Gaussian kernel evaluated at pixel coordinates. Note: The Gaussians must be sorted in front-to-back order, i.e by depth, to ensure correct alpha compositing.

The network is trained by minimizing the photometric reconstruction error between rendered and ground-truth images across all viewpoints. The photometric loss is defined as:

$$\mathcal{L}_{\text{photo}} = \sum_{k=1}^{K} \sum_{(u, v)} \| \boldsymbol{C}_k(u, v) - \boldsymbol{I}_k(u, v) \|_2^2 \tag{3}$$

where $\boldsymbol{I}_k(u, v)$ denotes the ground-truth color at pixel $(u, v)$ in image $I_k$. Additionally, excessively sharp ellipsoids may introduce glitch artifacts, compromising the visual coherence of the scene. To address these issues, we enhance the representation's robustness and visual fidelity by imposing a shape constraint during training. Inspired by Xie et al. (2023); Ling et al. (2023); Li et al. (2024), we introduce a regularization term that penalizes ellipsoidal eccentricity during the optimization of the 3D Gaussian scene:

$$\mathcal{L}_{\text{shape}} = \frac{1}{|\boldsymbol{G}|} \sum_{G_i \in \boldsymbol{G}} 1 - \frac{\min(s_i)^2}{\max(s_i)^2}, \tag{4}$$

Where $s_i$ represents the scaling along each axis. The overall loss function is

$$\mathcal{L}_{\text{total}} = \mathcal{L}_{\text{photo}} + \lambda \mathcal{L}_{\text{shape}} \tag{5}$$

### 3.2 PHYSICALLY-BASED ANALYSIS UNDER EXTERNAL FORCES

We model the dynamic behavior of each Gaussian primitive $\mathcal{G}_i$ under physically plausible forces. In particular, each primitive is assumed to be influenced by a user-defined external force field $\boldsymbol{F}_{\text{ext},i} \in \mathbb{R}^3$, which may arise from various vector fields simulating natural or stylized effects. Typical examples include:

- **Spiral Field**: Induces rotational motion around a predefined axis or center;

- **Wind Field**: Generates directional motion;

- **Oscillatory Field**: Temporally harmonic motion;

- **Noise Field**: Simulates stochastic turbulence.

To analyze the physical response of each Gaussian, we consider two characteristic regimes of force application:

1. **Instantaneous Impulse Response:** When a static Gaussian is suddenly subjected to an external force, its instantaneous velocity change is governed by Newton's second law in impulsive form:

$$\Delta \boldsymbol{v}_i = \frac{\boldsymbol{F}_{\text{ext},i} \cdot \Delta t}{m_i}, \tag{6}$$

where $\Delta \boldsymbol{v}_i$ is the velocity increment, $\Delta t$ is a small time interval, and $m_i$ is the mass of the Gaussian primitive (defined in later sections).

2. **Steady-State Motion under Continuous Force:** For sustained motion, we incorporate both external forces and internal structural forces to model the dynamic evolution. The second-order motion equation is:

$$m_i \cdot \frac{d^2 \boldsymbol{p}_i(t)}{dt^2} = \begin{cases} \boldsymbol{F}_{\text{ext},i}, & t = 0 \\ \boldsymbol{F}_{\text{ext},i} + \boldsymbol{F}_{\text{int},i}(t), & t > 0 \end{cases} \tag{7}$$

where:

- $\boldsymbol{F}_{\text{ext},i}$ is the constant or view-dependent external force;
- $\boldsymbol{F}_{\text{int},i}(t)$ represents internal forces, including elasticity and damping, from neighboring Gaussians.

We further decompose the internal force $\boldsymbol{F}_{\text{int},i}(t)$ into two components:

$$\boldsymbol{F}_{\text{int},i}(t) = \boldsymbol{F}_{\text{elastic},i}(t) + \boldsymbol{F}_{\text{damping},i}(t). \tag{8}$$

**Elastic Force.** Each Gaussian is associated with a rest position $\boldsymbol{p}_i^0$. The elastic force attempts to restore the primitive toward this rest position:

$$\boldsymbol{F}_{\text{elastic},i}(t) = -k_i \cdot \left( \boldsymbol{p}_i(t) - \boldsymbol{p}_i^0 \right), \tag{9}$$

where $k_i$ is a stiffness coefficient.

**Damping Force.** To prevent oscillatory behavior and simulate energy loss, a velocity-proportional damping term is added:

$$\boldsymbol{F}_{\text{damping},i}(t) = -\zeta_i \cdot \frac{d\boldsymbol{p}_i(t)}{dt}, \tag{10}$$

with $\zeta_i$ being the damping coefficient.

Putting all terms together, the motion of each Gaussian primitive follows a damped mass-spring system driven by external force:

$$m_i \cdot \frac{d^2 \boldsymbol{p}_i(t)}{dt^2} = \boldsymbol{F}_{\text{ext},i} - k_i \cdot \left( \boldsymbol{p}_i(t) - \boldsymbol{p}_i^0 \right) - \zeta_i \cdot \frac{d\boldsymbol{p}_i(t)}{dt}, \quad t > 0. \tag{11}$$

This physically-based dynamic model forms the core of our animation and deformation framework. It enables smooth, controllable motion propagation across the Gaussian field and supports realistic responses to both global and localized vector fields.

### 3.3 NEWTONIAN SIMULATION AND CINEMAGRAPH RENDERING

#### 3.3.1 3D GAUSSIAN MASS APPROXIMATION

A key aspect of physically-based simulation is assigning mass to each 3D Gaussian primitive. Since each primitive resembles a semi-transparent ellipsoid in 3D space, we estimate its mass from spatial extent. We model each Gaussian $\mathcal{G}_i$ as a spatial probability density function and compute a confidence ellipsoid enclosing a fixed proportion $\lambda \in (0, 1)$ of its mass. Let $\Sigma_i \in \mathbb{R}^{3 \times 3}$ be the covariance matrix, and $\chi_\lambda^2(3)$ the chi-squared quantile with 3 degrees of freedom. The ellipsoidal volume is:

$$V_i = \frac{4}{3}\pi \sqrt{\det(\Sigma_i)} \cdot \left( F_{\chi^2(3)}^{-1}(\lambda) \right)^{3/2} \tag{12}$$

Finally, we define the mass of the Gaussian as proportional to its volume via a global density parameter $\rho$: $m_i = \rho \cdot V_i$. By tuning $\rho$ and $\lambda$, users can control the absolute mass scale and the spatial confidence used for volume estimation, respectively.

#### 3.3.2 GAUSSIAN PRIMITIVES CLUSTERING

Although 3D Gaussian primitives provide detailed scene representations, they lack structural organization. In contrast, real-world objects often exhibit local geometric and dynamic coherence. To model this, we cluster nearby Gaussians with similar spatial and appearance features into coherent units, referred to as *SuperGaussians*. Following LoopGaussian Li et al. (2024), we adopt a

supervoxel-based clustering strategy. Let the full Gaussian set be $\mathbf{G} = \{G_i\}_{i=1}^N$, and the resulting SuperGaussians be $\mathbf{C} = \{C_k\}_{k=1}^K$, where each $C_k \subset \mathbf{G}$. Clustering is initialized by partitioning the scene into uniform voxels, selecting a seed Gaussian per voxel, and assigning others based on a composite distance metric. Each cluster shares physical attributes such as mass and motion. The total cluster mass is $M_k = \sum_{G_i \in C_k} m_i$ and the mass-weighted center of mass is:

$$\bar{\boldsymbol{\mu}}_k = \frac{1}{M_k} \sum_{G_i \in C_k} m_i \cdot \boldsymbol{p}_i. \tag{13}$$

Cluster motion follows Newton's law:

$$M_k \cdot \frac{d^2 \bar{\boldsymbol{\mu}}_k}{dt^2} = \sum_{G_i \in C_k} (\boldsymbol{F}_{\text{ext},i} + \boldsymbol{F}_{\text{int},i}). \tag{14}$$

We then propagate the cluster's motion to each constituent Gaussian:

$$\boldsymbol{p}_i(t + \Delta t) = \boldsymbol{p}_k(t) + \boldsymbol{v}_k(t) \cdot \Delta t + \delta_i(t), \tag{15}$$

where $\boldsymbol{v}_k(t)$ is the cluster velocity and $\delta_i(t)$ captures local deformation. This hierarchical abstraction ensures efficient, coherent simulation while preserving local motion flexibility.

### 3.3.3 LOOPABLE CINEMAGRAPH RENDERING.

To enable temporally seamless animations, we adopt a loopable rendering strategy inspired by Loop-Gaussian Li et al. (2024). The goal is to generate a continuous and smooth animation cycle by blending forward and backward simulation trajectories. Given the governing motion equations, we first simulate a forward sequence $\{F_f(t)\}_{t=0}^T$ using the standard position update rule (Eq. 7). Simultaneously, a backward sequence $\{F_b(t)\}_{t=0}^T$ is generated by applying time-reversed dynamics. To create a smooth transition and form a perfect loop, we linearly interpolate between the two trajectories:

$$F(t) = \beta(t) \cdot F_f(t) + (1 - \beta(t)) \cdot F_b(T - t), \quad 0 \le t \le T, \tag{16}$$

where the weight $\beta(t) = 1 - \frac{t}{T}$ gradually decays over time. This ensures a smooth handoff from forward to backward motion, yielding a cyclic sequence that returns seamlessly to the initial state.

## 3.4 STRUCTURAL OPTIMIZATION

### 3.4.1 INTRODUCTION OF RIGID BODY

To further enhance the structural realism of SuperGaussians, we introduce a rigid body simulation framework that models both translational and rotational dynamics at the cluster level. This design is motivated by the observation that, within each SuperGaussian, primitives often exhibit coherent motion patterns—making them suitable to be approximated as rigid bodies. Each cluster $C_j$ is characterized not only by its total mass $M_j$, center of mass $\bar{\boldsymbol{\mu}}_j$, and linear velocity $\boldsymbol{v}_j$, but also by rotational quantities including angular velocity $\boldsymbol{\omega}_j \in \mathbb{R}^3$, angular acceleration $\boldsymbol{\beta}_j \in \mathbb{R}^3$, and a time-varying moment of inertia $I_j(t)$. Let $\boldsymbol{r}_i(t) = \boldsymbol{p}_i(t) - \boldsymbol{\mu}_j(t)$ denote the displacement vector from the cluster center to Gaussian $G_i$. When a net torque $\boldsymbol{M}_{\text{total},j}(t)$ is applied to the cluster, the angular acceleration is given by:

$$\boldsymbol{\beta}_j(t) = \frac{\boldsymbol{M}_{\text{total},j}(t)}{I_j(t)}. \tag{17}$$

The angular velocity evolves similarly to eq.6.

Each Gaussian $G_i \in C_j$ is then rotated around the cluster center and translated along the linear trajectory. The updated position is:

$$\boldsymbol{p}_i(t + \Delta t) = \mathbf{R}(\boldsymbol{\omega}_j(t), \Delta t, \boldsymbol{r}_i(t)) + \boldsymbol{\mu}_j(t + \Delta t), \tag{18}$$

where $\mathbf{R}(\boldsymbol{\omega}, \Delta t, \boldsymbol{r}(t))$ is a rotation operator that applies a rotation to the vector $\boldsymbol{r}(t)$ pointing from the center of mass to the gaussian, of angle $\|\boldsymbol{\omega} \cdot \Delta t\|$ around the axis defined by $\boldsymbol{\omega}$, typically implemented via Rodrigues' formula. This dynamic rigid-body abstraction captures both translational and rotational effects in a physically coherent manner, enabling efficient and realistic simulation of structured Gaussian groups.

### 3.4.2 LOCAL STRUCTURAL OPTIMIZATION VIA CONSTRAINT GRAPH

To enable efficient and physically consistent internal force modeling, we introduce a sparse, locality-aware constraint graph that links only spatially relevant Gaussian pairs. Unlike naïve global formulations—which connect all primitives and incur high computational cost—our approach reflects real-world physical systems where forces act primarily between neighboring elements.

**Graph Construction.** To model internal forces efficiently, we construct a sparse constraint graph $\mathcal{B}$, connecting only spatially and semantically relevant Gaussian pairs.

**Gaussian-Level Construction.** For each Gaussian $G_i$, we find $K$-nearest neighbors $\mathcal{N}_i$ using a distance metric $D(\cdot, \cdot)$ that combines spatial and appearance similarity. We retain only those neighbors satisfying:

$$D(G_j, G_i) \leq \bar{D}_i, \quad \text{with } \bar{D}_i = \frac{1}{K} \sum_{G_j \in \mathcal{N}_i} D(G_j, G_i).$$

This results in a sparse, undirected graph $\mathcal{B}$ where edges reflect plausible internal forces.

**Extension to SuperGaussians.** We generalize the graph to cluster level by connecting SuperGaussians $C_i, C_j$ based on their mass centers: $D(C_i, C_j) = \|\bar{\boldsymbol{\mu}}_i - \bar{\boldsymbol{\mu}}_j\|$. Thresholding is applied as in the Gaussian-level case.

**Torque Computation.** For cluster $C_j$, the total torque is computed by summing contributions from constituent Gaussians:

$$\boldsymbol{M}_{\text{total},j}(t) = \sum_{G_i \in C_j} \left(\boldsymbol{p}_i(t) - \bar{\boldsymbol{\mu}}_j(t)\right) \times \boldsymbol{F}_{\text{ext},i}(t). \tag{19}$$

Internal forces defined by the constraint graph act symmetrically between Gaussian pairs and are centered around the mass center rather than being applied independently to individual Gaussians. As a result, their contributions cancel out in the torque computation, ensuring that they do not induce net rotational effects. This torque-neutral property enhances the physical stability of the simulation. This graph-based structure reduces computational complexity from quadratic to near-linear by limiting force computation to local neighborhoods, enhances simulation stability by avoiding noisy long-range interactions, and offers flexible control through adjustable connectivity.

## 4 EXPERIMENTS

### 4.1 DATASETS AND EXPERIMENTAL SETTINGS

We use two datasets. The first dataset is the NeRF synthetic dataset Mildenhall et al. (2021), which includes several high-quality static scenes. The other is ShapeSplatsV1 Ma et al. (2024); Chang et al. (2015), a large-scale dataset of Gaussian splats containing 65K objects across 87 unique categories. A unified time step $\Delta t$ is used across all updates involving position, velocity, angular velocity, and rotation. Structural forces are governed by two coefficients: the elastic coefficient $k$ and damping coefficient $\zeta$. We set the coefficients $k$ as: $k = C_k \cdot \min\left(\frac{1}{\Delta t^2}, \frac{1}{\Delta t}\right)$ and $\zeta$ as the same, where $C_k$ and $C_\zeta$ are tunable constants. See appendix for details about coefficients design motivation.

### 4.2 COMPARED WITH EXISTING APPROACHES

We conduct a comparative study that explores three existing methods (Figure 5)—— PhysGaussian Xie et al. (2023), PhysDreamer Zhang et al. (2024), and DreamGaussian4D Ren et al. (2023). Experiments show that our method robustly preserves physical plausibility across a wide range of object motions, delivering both compelling liveliness and high fidelity (See Appendix for more discussion).

### 4.3 EVALUATION UNDER DIVERSE FORCE FIELDS

We evaluate our framework under four distinct force fields: spiral, wind, oscillatory, and noise, as shown in Fig. 2 and detailed in Sec. 3.2. The spiral field induces rotational forces, while the wind field applies vertical shear; the oscillatory field introduces harmonic displacement, and the noise

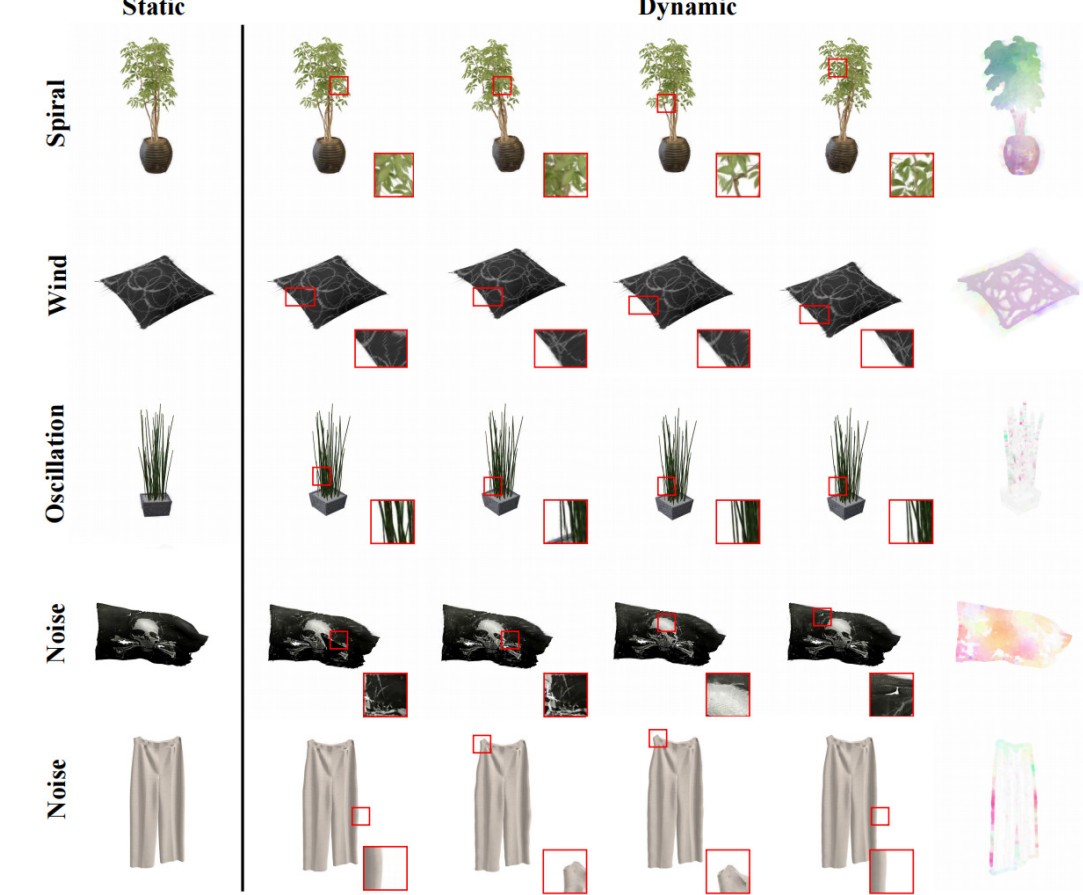

Figure 2: Visual results. Average optical flow visualizations (rightmost column) reveal that our method adapts to diverse force fields. Spiral and wind fields induce structured deformations in *Ficus* and *Pillow*, while oscillation and noise fields preserve the integrity of *Equisetum*, *Skull*, and *Pants*, validating stability under varied external dynamics.

field simulates stochastic disturbances using zero-mean Gaussian noise. These forces test different physical properties: spiral and wind assess torsional and shear resistance, while oscillation and noise evaluate stability against periodic and chaotic perturbations. Results confirm that our method preserves structural integrity under all scenarios. Although we employ rigid-body units in simulation, this does not imply strict rigidity of the modeled objects. Instead, by adjusting the resolution of the supervoxel clustering, we control the granularity of physical behavior: higher resolution yields smaller clusters, enabling fine-grained local deformation and non-rigid behavior.

## 4.4 ABLATION STUDY

In this section, we conduct several experiments to evaluate the effectiveness of our proposed method. The following text discusses two topics: Global vs. Local Constraints, and Structural force. Another topic about with clustering or not is presented in appendix.

***Global vs. Local Constraints***. In real-world physics, structural forces typically arise from local interactions—mass units exert force on nearby elements, while distant ones contribute minimally. To evaluate the validity of our locality-aware constraint model, we conduct simulations under both local and global constraints. As shown in Fig. 3, global constraints cause noticeable distortions: the Ficus under a spiral field and the pillow under a wind field both exhibit unrealistic deformations, while the skull under a noise field remains unstable. Notably, the lightflow map of the pillow reveals that global constraints disrupt its geometric consistency. In contrast, local constraints preserve structural coherence and produce more physically plausible results. Furthermore, local constraints

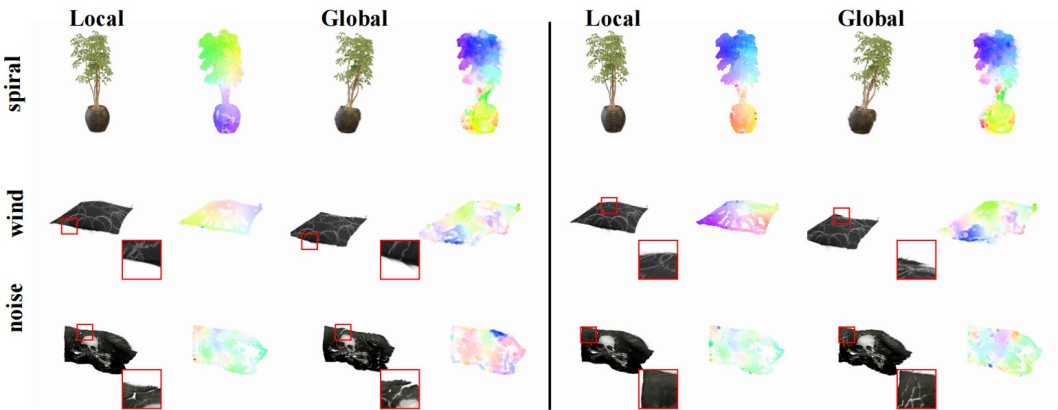

Figure 3: Global constraints indeed bring unrealistic distortion to the scenes. In the general case, mass units, which are far away, produce a negligible interaction force. If regarding their interaction as the same as neighboring mass units, the computed Lagrangian actually consists of large errors.

significantly reduce computational cost—the global constraint setup requires roughly three times longer to compute.

***Structural Force***. We further analyze the role of structural forces in clustered scenes under different configurations: with and without structural terms, and with partial versus full structural modeling. Results are shown in Fig. 4. Structural forces include elastic components, which constrain relative displacement between clusters, and damping components, which suppress relative motion intensity. In our experiment ($C_k = C_\varsigma = 0.2$), Fig. 4(c) shows that applying only elastic forces maintains relative cluster positions but fails to suppress local oscillations, leading to noticeable tearing. Conversely,

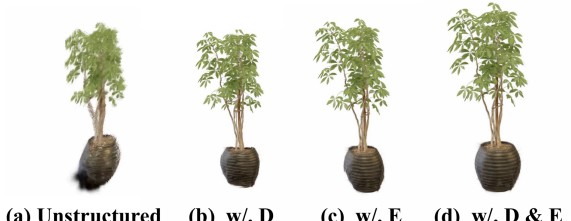

(a) Unstructured    (b) w/. D    (c) w/. E    (d) w/. D & E

Figure 4: Ablation study on structural force. Experimental results reveal that removing elastic forces leads to more severe tearing artifacts than removing damping. Elastic forces preserve inter-cluster spatial coherence, while damping suppresses excessive motion energy. "**D**" and "**E**" denote the presence of damping and elastic forces, respectively.

Fig. 4(b) shows that damping alone reduces motion intensity but fails to preserve structural alignment, worsening tearing over time. Visual comparisons suggest that elastic forces are more critical for preserving structural integrity.

## 5 CONCLUSIONS AND LIMITATIONS

We propose a physics-driven framework for 3D cinemagraph generation from multi-view images, addressing key limitations of prior heuristic or learning-based motion synthesis approaches. Our method leverages 3D Gaussian Splatting for scene reconstruction and embeds it within a physically grounded simulation environment. By incorporating external forces and internal structural constraints (including elasticity and damping), the system produces spatially coherent and physically plausible animations. It further enables interactive control via intuitive physical parameters, supporting fine-grained motion editing. Experiments demonstrate that our approach achieves high-fidelity, loopable cinemagraphs with enhanced realism, structural stability, and adaptability to diverse force fields, advancing interpretable and physically consistent dynamic scene generation. However, the framework still faces challenges in modeling complex or large-scale motions—such as exaggerated swinging or fluid-like dynamics—where realism and expressiveness remain limited.

## ETHICS STATEMENT

This work does not involve human subjects, animal experiments, or sensitive personal data. The datasets used (e.g., NeRF synthetic scenes and ShapeSplatsV1) are publicly available 3D reconstruction benchmarks that contain no personally identifiable information. Our method focuses on physics-based animation of static 3D scenes from multi-view images and does not generate harmful, deceptive, or privacy-invasive content. The proposed framework operates entirely on synthetic or publicly released geometric data and adheres to principles of fairness, transparency, and research integrity as outlined in the ICLR Code of Ethics. The authors declare no conflicts of interest.

## REPRODUCIBILITY STATEMENT

To support reproducibility, we provide the following: (1) Full implementation details—including 3D Gaussian reconstruction settings, force field configurations, clustering strategy, and simulation parameters—are described in Sections 4.1 and Appendix B and Appendix C. (2) The physical modeling equations (e.g., mass approximation, elastic/damping forces, torque computation) are explicitly derived in Appendix A. (3) All experiments use fixed random seeds, and quantitative results are averaged over multiple runs. While code is not included in the submission due to double-blind review, we commit to releasing anonymized implementation upon acceptance.

## LLM USAGE STATEMENT

Large Language Models (LLMs) were used in this work solely as a general-purpose writing assistance tool—for example, to improve grammar, clarify phrasing, or check technical terminology in the manuscript. LLMs did not contribute to the conception of the research idea, physical modeling, algorithm design, experimental setup, or interpretation of results. All scientific content, including equations, simulation frameworks, and claims, was developed and verified by the authors. No LLM was used to generate novel technical content or to draft substantial portions of the paper. As required by ICLR policy, we confirm that LLMs are not listed as authors, and we take full responsibility for all content under our names.

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

APPENDIX

## A  MOMENT OF INERTIA DERIVATION FOR SUPERGAUSSIAN CLUSTERS

To simulate angular acceleration for a SuperGaussian cluster $C_j$ under Newtonian dynamics, we require an accurate estimation of the instantaneous moment of inertia around a given axis.

In the rigid body abstraction introduced for each SuperGaussian cluster $C_j$, we must account for both translational and rotational dynamics. The rotational behavior depends on the cluster's **instantaneous moment of inertia** $I_j(t)$, which varies over time as the internal spatial configuration evolves.

Let $\boldsymbol{k}_j \in \mathbb{R}^3$ be a unit vector indicating the instantaneous axis of rotation (typically aligned with the angular velocity $\boldsymbol{\omega}_j$). For each Gaussian $G_i \in C_j$, let $\boldsymbol{r}_i(t) = \boldsymbol{p}_i(t) - \bar{\boldsymbol{\mu}}_j(t)$ denote its displacement from the cluster's center of mass at time $t$. Then, the **scalar moment of inertia** around axis $\boldsymbol{k}_j$ is given by:

$$I_j(t) = \sum_{G_i \in C_j} m_i \cdot \left( \|\boldsymbol{r}_i(t)\|^2 - (\boldsymbol{r}_i(t) \cdot \boldsymbol{k}_j)^2 \right), \tag{20}$$

This expression is derived from the general moment of inertia formula for point masses rotating about an arbitrary axis, and measures the effective rotational resistance of the Gaussian cluster. Intuitively, it sums the mass-weighted squared perpendicular distances from each Gaussian to the axis of rotation.

The term $\|\boldsymbol{r}_i(t)\|^2 - (\boldsymbol{r}_i(t) \cdot \boldsymbol{k}_j)^2$ computes the squared length of the component of $\boldsymbol{r}_i(t)$ orthogonal to $\boldsymbol{k}_j$, i.e., the squared radius of rotation for each primitive.

Because both $\boldsymbol{r}_i(t)$ and $\boldsymbol{k}_j$ may evolve due to deformation and force interaction, $I_j(t)$ must be re-evaluated at each simulation step. This time-varying inertia enables accurate computation of angular acceleration under applied torques using:

$$\boldsymbol{\beta}_j(t) = \frac{\boldsymbol{M}_{\text{total},j}(t)}{I_j(t)}, \tag{21}$$

Combined with the translational dynamics governed by Newton's law, this formulation completes the rigid-body modeling for each SuperGaussian, supporting physically realistic and stable rotation-aware animation.

**Remark.**  In physics-augmented 3D Gaussian Splatting, the rotational behavior of a Gaussian cluster is jointly determined by its mass distribution and geometric configuration. Unlike rigid meshes, Gaussian primitives possess adaptive spatial spreads and dynamic centroids, causing the inertia properties to evolve continuously during simulation. A precise inertia formulation is thus required to couple mass-weighted geometry with rotational dynamics, enabling accurate torque response, stability control, and motion coherence. We begin with the classical scalar inertia definition for a fixed rotation axis, reformulate it using vector projections for efficient per-step updates, and finally generalize to a full inertia tensor that accommodates arbitrary-axis rotations and supports advanced analyses such as torque decomposition and principal axis alignment.

**General Formulation.**  Given a rotation axis represented by a unit vector $\boldsymbol{k}_j \in \mathbb{R}^3$, the scalar moment of inertia $I_j(t)$ of cluster $C_j$ at time $t$ is defined as the sum of mass-weighted squared distances from each point mass $G_i \in C_j$ to the axis:

$$I_j(t) = \sum_{G_i \in C_j} m_i \cdot d_i^2, \tag{22}$$

where $d_i$ denotes the perpendicular distance from the $i$-th point to the axis defined by $\boldsymbol{k}_j$.

**Vector Expansion.**  The perpendicular distance $d_i$ can be written using projection properties:

$$d_i^2 = \|\boldsymbol{r}_i(t)\|^2 - (\boldsymbol{r}_i(t) \cdot \boldsymbol{k}_j)^2,$$

where $\boldsymbol{r}_i(t) = \boldsymbol{p}_i(t) - \boldsymbol{\mu}_j(t)$ is the position of Gaussian $G_i$ relative to the cluster centroid $\boldsymbol{\mu}_j(t)$.

Substituting into the inertia definition, we obtain:

$$I_j(t) = \sum_{G_i \in C_j} m_i \left( \|\boldsymbol{r}_i(t)\|^2 - (\boldsymbol{r}_i(t) \cdot \boldsymbol{k}_j)^2 \right). \tag{23}$$

This formulation captures the dynamic configuration of the cluster and must be updated at each simulation step.

**Tensor Generalization**  For more general cases where the rotation axis changes, the moment of inertia tensor $\mathbf{I}_j(t)$ can be computed as:

$$\mathbf{I}_j(t) = \sum_{G_i \in C_j} m_i \left( \|\boldsymbol{r}_i(t)\|^2 \cdot \mathbf{I}_3 - \boldsymbol{r}_i(t)\boldsymbol{r}_i(t)^\top \right),$$

and the scalar inertia around axis $\boldsymbol{k}_j$ is:

$$I_j(t) = \boldsymbol{k}_j^\top \mathbf{I}_j(t)\boldsymbol{k}_j.$$

This tensor form allows for flexible handling of arbitrary rotation axes and supports future extensions such as torque decomposition or eigen-analysis.

## B  DATASET

To evaluate our method, we employ two synthetic datasets selected for their clean backgrounds and controllable object properties, which facilitate the isolation and observation of motion behaviors under physics-based simulation. Using synthetic datasets avoids the visual interference caused by cluttered real-world scenes and enables more precise tracking of object dynamics.

The first dataset is the **NeRF Synthetic dataset** Mildenhall et al. (2021), which contains multiple high-quality static scenes rendered with photorealistic geometry and texture details. This dataset provides well-structured spatial information, making it suitable for controlled simulation studies.

The second dataset is ShapeSplatsV1 Ma et al. (2024); Chang et al. (2015), a large-scale collection of 3D Gaussian splats comprising approximately 65,000 objects across 87 categories. It is derived from ShapeNetCore, ShapeNet-Part, and ModelNet, with the majority (about 52,000 objects in 55 categories) sourced from ShapeNetCore. Each object is stored in PLY format, where Gaussian attributes—including position, color, opacity, and orientation—are encoded as custom vertex attributes. This rich attribute representation enables direct integration into Gaussian-based rendering and simulation pipelines.

## C  EXPERIMENTAL SETTINGS

All experiments are conducted on a single NVIDIA GeForce RTX 4090 using the PyTorch framework Paszke et al. (2019). The force field is estimated by a two-layer MLP with hidden dimensions of 128 and 64, where positional encoding is applied to the input. For final rendering, we follow the settings of LoopGaussian Li et al. (2024), generating videos with a resolution of $900{\times}900$ over 48 frames. One scene in the dataset typically has 100 pictures of different views.

Since each Gaussian is updated based on Newtonian dynamics, we aim to bound the total displacement over two time steps:

$$\Delta\mathbf{p} = \mathbf{p}_{t+2} - \mathbf{p}_t = 2\mathbf{v}_t\Delta t + \frac{\boldsymbol{F}_{\text{total}}}{m}\Delta t^2 \leq O(C),$$

where $\boldsymbol{F}_{\text{total}}$ denotes the aggregated force. Assuming internal forces dominate at each step, we approximate:

$$\|\boldsymbol{F}_{\text{total}}\| \approx \|\boldsymbol{F}_{\text{int}}\| = k \cdot \Delta x + \zeta \cdot \Delta v,$$

which simplifies to:

$$\|\boldsymbol{F}_{\text{total}}\| \approx k \cdot \Delta t^2 + \zeta \cdot \Delta t.$$

To maintain numerical stability and avoid overshooting, we enforce:

$$k \leq O(\Delta t^{-2}), \quad \zeta \leq O(\Delta t^{-2}).$$

Additionally, to constrain velocity change:

$$\Delta \mathbf{v} = \mathbf{v}_{t+1} - \mathbf{v}_t = \frac{\boldsymbol{F}_{\text{total}}}{m} \cdot \Delta t \approx k \cdot \Delta t + \zeta \cdot \Delta t.$$

We therefore set the coefficients as:

$$k = C_k \cdot \min\left(\frac{1}{\Delta t^2}, \frac{1}{\Delta t}\right),$$

$$\zeta = C_\zeta \cdot \min\left(\frac{1}{\Delta t^2}, \frac{1}{\Delta t}\right), \tag{24}$$

Smaller stiffness and damping coefficients generally yield more rigid and numerically stable structures. In our implementation, we empirically set:

$$\Delta t = 0.1, \quad C_k = C_\zeta = 0.01,$$

which achieves a favorable trade-off between motion controllability, simulation stability, and structural coherence.

The overall parameterization of our system remains compact, comprising only the structural force coefficients $(C_k, C_\zeta)$, a density parameter $\rho$, and an external force field. Default values are provided for the structural and density parameters, thereby reducing the burden on the user. In practice, the user only needs to supply the external force field, specified as force vectors at selected spatial positions.

## D  ADDITIONAL EVALUATION

We present our comparison evaluation in Figure 5. The methods used to be compared are PhysGaussian Xie et al. (2023), PhysDreamer Zhang et al. (2024), and DreamGaussian4D Ren et al. (2023). PhysGaussian, a cornerstone of 3D-GS physical simulation, achieves convincing motion for soft-body materials thanks to its MPM-based pipeline; however, it still suffers from implausible structural tearing when handling moderately rigid objects or stronger perturbations. PhysDreamer borrows heavily from PhysGaussian and on soft bodies, likewise exhibits artifacts such as volume non-conservation, excessive stretching, and over-bending, all of which degrade the realism of the simulation. DreamGaussian4D, in essence, builds upon Stable-Video-Diffusion, using a diffusion model to guide the generation of physically simulated animations. The result in the figures seems to be disordered and aliased. Because the resulting videos are not subject to direct physical constraints, they appear unrealis-

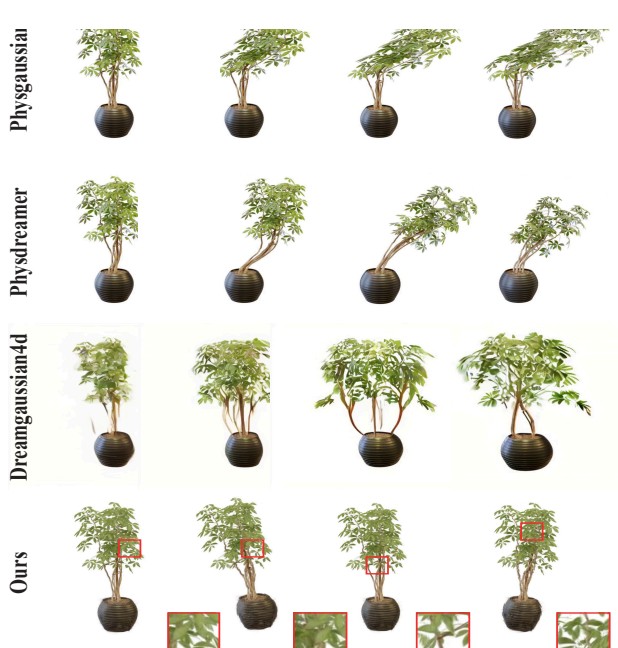

Figure 5: Qualitative comparison of our PBS simulation methods. Our approach stably preserves scene geometry and structural integrity, whereas existing 3D-GS physics-based and diffusion-based methods—despite offering some visual liveliness—often fail to maintain physically consistent scene structures.

tic and even lose a degree of vividness. In contrast, our method robustly preserves physical plausibility across a wide range of object motions, delivering both compelling liveliness and high fidelity. The loopable rendering further enriches motion diversity without sacrificing realism.

The ficus scene is chose from NeRF-synthetic dataset Mildenhall et al. (2021), and the equisetum scene is chose from ShapeSplatV1 Ma et al. (2024); Chang et al. (2015).

Another concrete ablation result is presented in Figure 6, where the effect of simulating with or without clustering is explored. Without structural forces, simulating individual Gaussians leads to loose and incoherent motion under a wind field. In contrast, clustering introduces strong local coherence by implicitly imposing structural constraints, so that Gaussians within

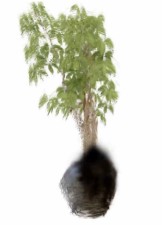 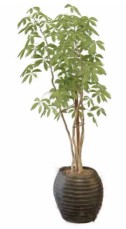

(a) **Unstructured**  (b) **Unstructured w/ clustering**

Figure 6: Ablation results of cluster.

a cluster move collectively, thereby reducing tearing artifacts. As illustrated in the figure, both frames are sampled from the same third frame of the video, clearly showing that clustered simulation preserves more stable structures at the beginning, highlighting the critical role of simulation-unit granularity in determining overall stability.

