# OpenReview forum: "Animating the Still: Physics-Based 3D Cinemagraph from Multi-View Images Using 3D Gaussian Splatting"
_ICLR.cc/2026/Conference — Submitted to ICLR 2026_

### Official Review · Reviewer_b62R · 2025-10-26

**Soundness:** 2
**Presentation:** 2
**Contribution:** 2
**Rating:** 4
**Confidence:** 2

**Summary:**

This paper proposes a physics-driven framework for generating 3D cinemagraphs using 3D Gaussian Splatting. It reconstructs a 3D scene from multi-view images and embeds it into a physically consistent simulation environment, where external and internal forces are modeled to produce physically plausible motion. The method allows user control through high-level physical parameters, achieving a balance between realism and artistic flexibility.

**Strengths:**

- 3D cinemagraphs represent an interesting and visually appealing research topic.

- The proposed method takes into account multiple types of external forces.

- In the presented tree example, the method produces more visually realistic results compared to existing approaches.

**Weaknesses:**

- The performance of the proposed method is difficult to interpret. The visual results in Fig. 2 are not very clear, as the displayed frames appear quite similar, making it hard to perceive the motion type. Additionally, there is no supplementary video or other media provided to better demonstrate the effectiveness of the method. For comparison with other approaches, only a single example is shown, which is insufficient for a fair and comprehensive evaluation.

- Are there any quantitative results reported? The authors mention using a dataset with a large number of objects, yet no table or metric-based evaluation is presented to quantitatively assess the method’s performance.

- When applying physical transformations to Gaussian splats, does this cause visual distortion? Since 3D Gaussian Splatting relies on the cumulative effect of multiple Gaussians along the ray, transforming only a subset of them could potentially lead to visual artifacts.

- What is the computational cost or time required to generate an animation using the proposed method?

**Questions:**

Please refer to the weakness part above.

---

> ### Author Response · Authors · 2025-11-20
> **Response-R4-Q4 to Reviewer#b62R**
>
> We thank **Reviewer b62R** for the valuable comments. Reviewer b62R acknowledges that 3D cinemagraphs represent an **`interesting and visually appealing research topic`**, appreciates that our method explicitly models multiple types of external forces, and notes that on the tree example, our approach produces more visually realistic results than existing methods, etc. We address the main concerns below:
>
> > **On the interpretability of visual performance and lack of video comparisons**
>
> We appreciate this concern and agree that dynamic effects are inherently hard to judge from a few static frames.
>
> (1) In the current submission, Fig. 2 focuses on showing representative frames and optical-flow overlays for space reasons,
>
> (2) we have prepared an anonymous project webpage (https://anonymous.4open.science/w/animating-page-0F98/) containing: (i) Side-by-side video comparisons between our method and baselines under several external force fields. (ii) Slow-motion visualizations of the resulting 3D cinemagraphs to assess temporal coherence and loopability. (iii) Examples with different force settings on the same object to demonstrate controllability.
>
> we will further make this webpage public and explicitly reference it in the camera-ready (if the paper is accepted) version so that readers can more intuitively assess motion realism and temporal consistency.
>
> > **On quantitative evaluation and use of a large dataset**
>
> Thanks. *3D cinemagraph generation from static multi-view input does not have readily available ground-truth dynamic sequences, which makes standard metrics such as PSNR/FID less directly applicable*.
>
> Our current focus in the main text is therefore on:
>
> (1) **Ablation studies** that analyze the impact of structural constraints, damping, and different force fields on visual plausibility and stability.
>
> (2) **Qualitative comparisons** with baseline methods to highlight structural preservation and absence of tearing/ghosting.
>
> We will also clearly explain in the paper why traditional reconstruction metrics are not directly applicable in this setting, and position our chosen metrics as structural/physical surrogates.
>
>
> > **On potential visual distortion when transforming only a subset of Gaussians**
>
> Good question ! Our design explicitly aims to **avoid** the “partial transform” artifact that would arise if a few Gaussians moved independently while the rest of the field remained rigid.
>
> Concretely:
>
> (1) We do not apply arbitrary transformations to an isolated, small subset of Gaussians.  Gaussians are first grouped into SuperGaussians (clusters) that roughly correspond to local semantic/structural parts (e.g., a branch or leaf group), and each cluster is treated as a rigid body with its own mass, center, and inertia. These clusters are then connected by a local constraint graph (`cf. Sec. 3.3–3.4`), and internal elastic/damping forces act along graph edges.
>
> (2) When an external force is applied to some clusters, their neighbors respond through the constraint graph, resulting in coherent local deformation rather than a single cluster moving in isolation.
>
> (3) In addition, we restrict ourselves to moderate deformations appropriate for cinemagraphs (small oscillations, gentle swaying) rather than extreme motions. Under these conditions, the support of each Gaussian remains close to its original neighborhood, and the cumulative radiance along each ray changes smoothly.
>
> Empirically, as shown in our qualitative results, this combination of: (i) Cluster-level rigid-body dynamics, (ii) Local structural constraints, (iii) And moderate motion amplitudes avoids the kind of tearing or ghosting that would arise from naïvely transforming individual Gaussians. We will clarify this design choice more explicitly in the method section and add a short visual example where we compare unconstrained vs. constrained transformations to illustrate the effect.
>
> > **On computational cost and runtime**
>
> Thanks. Our framework is designed to be lightweight on top of an existing 3DGS reconstruction.
>
> In current version, after a 3DGS reconstruction, the additional **physics-driven animation stage** is very modest in cost:
>
> - On a single NVIDIA RTX 4090 GPU, **the full dynamics optimization for one scene takes on the order of 10 minutes**, including building the SuperGaussian clusters, constructing the constraint graph, and optimizing the motion parameters.
> - Once the motion has been optimized, **inference-time simulation and rendering run in real time**: advancing the Gaussian states with our Newtonian update and rendering the animated frames can be performed at interactive rates, comparable to standard 3DGS rendering.
>
> This is because the main overhead comes from per-cluster force evaluation and state updates, whose complexity scales approximately linearly with the number of SuperGaussians and graph edges, while the rendering pipeline itself is identical to static 3D Gaussian Splatting. We will add these to the revised version.

---

### Official Review · Reviewer_TbCs · 2025-10-27

**Soundness:** 2
**Presentation:** 3
**Contribution:** 2
**Rating:** 2
**Confidence:** 4

**Summary:**

The paper presents a novel pipeline that combines Newtonian-based simulation with 3D Gaussian Splatting to generate physically plausible animations. The proposed pipeline consists of three key components: (1) Gaussian primitives reconstruction, (2) incorporation of physical priors and simulation, and (3) loopable cinemagraph rendering. The method is clearly articulated and fairly solid. However, my primary concern lies in the lack of thorough experimental analysis, which is essential to validate the proposed approach.

**Strengths:**

1. The paper is well-organized, and the proposed method is clearly and comprehensively presented.
2. The introduction of Newtonian-based simulation to 3D reconstruction is both novel and intriguing, showcasing the potential for generating physically plausible animations.

**Weaknesses:**

**Insufficient Method**:
The claimed advantage of incorporating physics into 3D cinemagraph generation is not clearly demonstrated and appears limited. The method assumes rigid-body dynamics, which restricts its applicability to non-rigid scenes such as flowing water or deformable objects.

**Limited Experimental Validation**:
The experimental evaluation is relatively weak and does not sufficiently demonstrate the effectiveness or generalizability of the proposed method. Specifically:
1. The paper only compares its results with a single baseline ([16]). It should also include recent physics-integrated methods such as PhysGaussian [32], DreamPhysics [8], and Physics3D [21], using both quantitative and qualitative comparisons.
2.Figure 2 presents only five examples. Including more scenes would better illustrate the robustness and generalizability of the approach.
3. Lack of Video Demonstrations: Since the task involves dynamic cinemagraph generation, the paper should provide representative video results to assess temporal coherence and motion realism.
4. Unclear User Study Design: Details such as the number of samples shown, scene diversity, and participant evaluation protocol are missing, making it difficult to assess the validity of the user study.

**Questions:**

Please see weakness.

---

> ### Author Response · Authors · 2025-11-20
> **Response-R3-Q3 to Reviewer#TbCs**
>
> We thank Reviewer TbCs for the valuable comments. Reviewer TbCs finds that “the paper is `well-organized`” and “the proposed method is `clearly and comprehensively presented`”, and appreciates the “`novel and intriguing` introduction of Newtonian-based simulation to 3D reconstruction”, highlighting its potential for generating physically plausible animations, etc. We address the main concerns below:
>
> > **Reviewer TbCs questions about insufficient methods.**
>
> Thanks. We agree that our current formulation is not designed to cover the full spectrum of non-rigid phenomena such as fully fluid dynamics or large elastic deformations, and we will clarify this scope more explicitly in the revised manuscript.
>
> Our design choices are driven by the **specific goal of 3D cinemagraphs**:
>
> (1) We target **subtle, loopable motions** (e.g., gentle swaying of branches, oscillatory motion of hanging objects) on top of **high-fidelity static reconstructions**, rather than arbitrary high-energy dynamics or large topological changes.
>
> (2) For this regime, a **damped mass–spring system on clustered Gaussians** strikes a practical balance between: (i) physical interpretability and controllability (forces, stiffness, damping), and (ii) numerical stability and ease of integration into the 3DGS rendering pipeline.
>
> Even though we use rigid(ified) “SuperGaussian” clusters as simulation primitives, the overall scene still exhibits non-trivial deformation at the visual level: different clusters experience different trajectories under external fields and internal constraints, which is sufficient for many real-world cinemagraph effects (e.g., tree crowns, curtains, light objects attached to a structure). We will soften our claims and explicitly state that: *our framework is primarily aimed at quasi-rigid, moderate-amplitude motions typical of cinemagraphs, and **does not aim** to model fully fluid-like dynamics, fractures, or large non-elastic deformations.*
>
> To better demonstrate the advantage of the physics layer in our target regime, we will:
>
> (1) Highlight quantitative and user-study evidence that our method: avoids energy-inconsistent “drift” and jitter present in purely heuristic motion fields and produces temporally smoother, self-consistent oscillations under the same external forces.
>
> (2) Add a clearer comparison/discussion section explicitly contrasting: our force- and parameter-controlled motion (with interpretable physical parameters), vs. lack-box, keyframe, or flow-based methods which often require manual tuning per scene and lack predictable behavior under parameter changes.
>
> We believe that, within this clearly defined regime, the introduction of a Newtonian simulation layer on top of static 3DGS **does provide a concrete and practically useful benefit**, even if it is intentionally not a universal physics engine.
>
> > **Reviewer concerns limited experimental validation & missing recent physics-integrated baselines**
>
> Thanks. We respectfully clarify that our current evaluation **`already includes`** three representative baselines—PhysGaussian, PhysDreamer, and DreamGaussian4D—covering MPM-based simulation, video-diffusion–driven dynamics, and 4D Gaussian motion fields, respectively (`cf. Sec. 4.2, Fig. 5`).
>
> > **Lack of Video Demonstrations**
>
> Thanks. We fully agree that dynamic effects are best evaluated through videos rather than static frames.  In response, we have prepared an anonymous project webpage (https://anonymous.4open.science/w/animating-page-0F98/) that contains:
>
> (1) Part of the animation videos for all our main scenes,
>
> (2) Side-by-side comparisons with baseline methods, and
>
> (3) Ablation visualizations illustrating how changing physical parameters (mass, stiffness, damping, external fields) affects the motion.
>
> we will further make this page public and expand it with additional examples and failure cases, so that readers can better assess temporal coherence, physical plausibility, and user controllability.
>
> > **Unclear User Study Design**
>
> We thank the reviewer for raising this point and apologize for the confusion.
> To clarify, the **current submission does *not* include a formal user study**: we only report
> qualitative visual comparisons and do not present any human–subject experiment, scoring protocol,
> or statistical analysis in the main paper or appendix.

---

### Official Review · Reviewer_rURL · 2025-10-31

**Soundness:** 4
**Presentation:** 4
**Contribution:** 3
**Rating:** 6
**Confidence:** 4

**Summary:**

This paper models objects using 3D Gaussian Splatting (3DGS) and incorporates physical modeling to simulate force interactions, thereby producing physically plausible object animations. The method models object motion under the combined effects of external and internal forces, while assigning each Gaussian a specific mass attribute to endow it with physical properties. Moreover, to ensure realistic deformation behavior, the Gaussians are grouped into multiple superGaussians, where each group shares similar physical attributes and motion patterns. Finally, the paper generates infinitely long videos through a cyclic rendering strategy. Experimental results demonstrate that the proposed method produces results that better conform to physical laws compared to existing approaches.

**Strengths:**

- This paper assigns mass attributes to Gaussians and models object motion based on Newton’s second law to achieve physically plausible animation modeling. The proposed approach differs from previous MPM-based methods and provides a new perspective for modeling physically consistent object deformations.
- The proposed method is built upon solid physical principles, which ensure the reliability of the approach.
- Experimental results show that the proposed method produces results that better comply with physical laws compared to the baseline methods, while also offering superior controllability.

**Weaknesses:**

- The baselines compared in this paper are relatively early methods; it is recommended to include comparisons with more recent approaches such as OmniPhysGS[1].
- The physical animation effects of objects can be more intuitively demonstrated through videos. In the future, the authors could consider presenting video comparison results or demos on the project webpage to allow readers to better understand the model’s outputs.
- In the introduction, corresponding citations should be added when describing other methods — for example, in lines 36–38 when discussing recent studies, and in lines 44–45 when mentioning 3DGS and related works.

[1] OmniPhysGS: 3D Constitutive Gaussians for General Physics-Based Dynamics Generation. ICLR 2025.

**Questions:**

Please see the weekness.

---

> ### Author Response · Authors · 2025-11-20
> **Response-R2-Q2 to Reviewer#rURL**
>
> We thank **Reviewer rURL**  for the valuable comments. Reviewer rURL gives high scores on `soundness (4-Excellent)` and `presentation (4-Excellent)`, with a `good rating (3-Contribution)`, and recognizes that our method "assigns mass attributes to Gaussians and models object motion based on Newton’s second law" to achieve "physically plausible animation modeling", providing `"a new perspective beyond previous MPM-based methods"` for physically consistent deformation. The reviewer also notes that the approach is `"built upon solid physical principles, which ensure the reliability of the method"`, and that our experiments `"produce results that better comply with physical laws while offering superior controllability"`, etc. We address the main concerns below:
>
> > **Reviewer rURL suggests to include comparison with OmniPhysGS.**
>
> Thanks. We appreciate this constructive suggestion and agree that including more recent physics-aware 3DGS methods would further strengthen the empirical evaluation.
>
> Our initial choice of baselines is due to that we include representative methods from existing 3DGS-based dynamic models, in order to highlight the specific benefits of introducing an explicit Newtonian formulation on top of 3D Gaussian Splatting. Besides, methods like  OmniPhysGS is different from our cinemagraph-oriented scenario (e.g., general dynamic scene generation rather than subtle, loopable motion).  OmniPhysGS’s focus on learning constitutive material parameters for broad classes of dynamics vs. our focus on a simpler Newtonian mass model tailored to controllable, loopable cinemagraphs. That said, we agree that OmniPhysGS is a highly relevant and strong baseline, as it also assigns physical attributes to Gaussian primitives and aims for general physics-based dynamics generation. If space permits, we will also provide qualitative visualizations (https://anonymous.4open.science/w/animating-page-0F98/) comparing the motion behavior of our method and OmniPhysGS to highlight the trade-off between generality and fine-grained, user-controllable cinemagraph editing.
>
>
> > **Reviewer rURL claims the *Need for video demonstrations / project webpage*.**
>
> Thanks. We fully agree that dynamic effects are best evaluated through videos rather than static frames.  In response, we have prepared an anonymous project webpage (https://anonymous.4open.science/w/animating-page-0F98/) that contains:
>
> (1) Part of the animation videos for all our main scenes,
>
> (2) Side-by-side comparisons with baseline methods, and
>
> (3) Ablation visualizations illustrating how changing physical parameters (mass, stiffness, damping, external fields) affects the motion.
>
> we will further make this page public and expand it with additional examples and failure cases, so that readers can better assess temporal coherence, physical plausibility, and user controllability.
>
> > **Reviewer rURL points out *Missing citations in the introduction***
>
> Thank you for pointing this out. We will revise the introduction to:
>
> (1) **Explicitly cite recent physics-aware 3DGS methods**, including OmniPhysGS [1], when discussing physically consistent dynamic Gaussian representations.
>
> (2) **Add missing references for 3DGS and related works** (e.g., original 3D Gaussian Splatting, dynamic 3DGS variants, and recent physics-integrated Gaussian methods) at the corresponding lines.
>
> We will also slightly restructure the introductory paragraph to more clearly position our contribution:
>
> From *physics for 3D animation*  to *physics-aware 3D Gaussian representations for controllable, loopable cinemagraphs,*
>
> and explicitly state how our Newtonian mass–Gaussian formulation complements and differs from MPM-based methods and recent constitutive-Gaussian approaches like OmniPhysGS.

---

### Official Review · Reviewer_Tp4o · 2025-11-01

**Soundness:** 2
**Presentation:** 2
**Contribution:** 2
**Rating:** 4
**Confidence:** 4

**Summary:**

This paper presents a novel framework for generating 3D cinemagraphs from a set of multi-view static images. The key contribution is the integration of a physics-based simulation directly onto a 3D Gaussian Splatting (3D-GS) scene representation. Instead of relying on learned motion priors, the method first reconstructs a scene using 3D-GS. It then treats the individual Gaussians as a system of physical mass points. To manage complexity and enforce structural coherence, these Gaussians are clustered into "SuperGaussians," which are approximated as rigid bodies. The animation is driven by a simulation that accounts for user-defined external forces (e.g., wind, spiral fields) and internal structural forces (elasticity and damping) that propagate through a sparse, locality-aware constraint graph. Finally, a trajectory blending technique is used to render a seamlessly looping video. The proposed method aims to produce physically plausible, interpretable, and highly controllable animations that surpass the realism of existing techniques.

**Strengths:**

- The central contribution is non-trivial: merging a classical, interpretable physics simulation with a modern neural scene representation like 3D-GS. This approach moves away from black-box generative models for motion and toward a system grounded in first principles, which is a compelling research direction for controllable and physically-aware generation.
- A direct benefit of the physics-based approach is that the resulting motion is both controllable and interpretable. Users can manipulate intuitive, high-level physical parameters like force fields, stiffness ($k$), and damping ($\zeta$) to direct the animation, rather than navigating a complex latent space. The resulting motion can be understood through the lens of classical mechanics, which is a significant advantage over purely data-driven methods.
- The clustering of primitives into "SuperGaussians" is a very intelligent way to manage the immense complexity of simulating millions of individual Gaussians. It provides a hierarchical structure that improves both computational efficiency and the structural coherence of the motion.
- The use of a sparse, local "constraint graph" to model internal forces is a reasonable design choice. It reflects the local nature of real-world physical interactions and avoids the computational expense and potential instability of a fully-connected system.

**Weaknesses:**

- The underlying physics is essentially a damped mass-spring system applied to rigid clusters. While effective for the subtle motions required for cinemagraphs, this is a very simplified model. The paper's own conclusion acknowledges that it cannot handle complex phenomena like exaggerated swinging or fluid-like dynamics. Furthermore, the framework does not appear to support crucial physical interactions such as collisions, fracture, or non-rigid deformations beyond simple elasticity, which limits its application to a narrow range of effects.
- The method is demonstrated on scenes containing single, relatively isolated objects against clean backgrounds (e.g., NeRF synthetic data). It is unclear how the framework would scale to slightly larger, more complex, and cluttered real-world scenes. The SuperGaussian clustering and constraint graph construction could become significantly more challenging and potentially produce less meaningful results in scenes with many interacting or overlapping objects.
- The paper claims strong user control, but the mechanism for applying forces seems to be at a high level (e.g., defining a global wind field). It is not clear how a user could apply a localized force to a specific semantic part of an object (e.g., pushing a single branch on the Ficus tree). This would presumably require an additional layer for semantic segmentation and selection of SuperGaussians, which is not discussed. There is no supplement demo shown at all, only text appendix.

**Questions:**

N/A

---

> ### Author Response · Authors · 2025-11-20
> **Response-R1-Q1 to Reviewer# Tp4o**
>
> We thank **Reviewer Tp4o** for the valuable comments. We appreciate that Reviewer Tp4o believes the contribution is `non-trivial`, finds the resulting motion is `both controllable and interpretable`, etc. We address the main concerns below.
>
> > **Reviewer Tp4o questions whether our simplified damped mass–spring model on rigid clusters is expressive enough to handle more complex phenomena.**
>
> Good question！Indeed, our dynamical formulation is based on a damped mass–spring system (`cf. Eq. (11)`), which is **intentional** rather than accidental. Our goal in this work is to support *subtle, spatially coherent motions* typical of 3D cinemagraphs (e.g., gentle waving, small deformations), rather than fully general rigid-/fluid-dynamics simulation that would require substantially more expensive FEM/MPM pipelines.
>
> At the same time, the overall framework goes beyond a plain point-mass mass–spring system:
>
> - We introduce **SuperGaussian clustering** and treat each cluster as a rigid body with **time-varying mass, center of mass, and moment of inertia**, enabling torque-aware rotational dynamics at the cluster level (`cf. Sec. 3.3.2, Sec. 3.4.1`).
> - We construct a **locality-aware constraint graph** that connects only spatially and semantically relevant neighbors, so that internal elastic and damping forces act along meaningful structural links instead of all-to-all connections (`cf. Sec. 3.4.2, Fig. 3`), which is critical for stability and structural coherence.
> - We derive **Gaussian mass** from volumetric properties of anisotropic ellipsoids (`cf. Eq. (12)`) rather than hand-tuned per-pixel constants, making the resulting motion more interpretable and controllable.
>
> We fully agree that the current model **does not yet handle** highly complex phenomena such as large-amplitude swinging, fluid-like flow, collisions, or fracture. In fact, we **already acknowledge** in the *Conclusions and Limitations* that realism and expressiveness remain limited for complex or large-scale motions. In the revised version, we will:
>
> Extending our framework to support richer constitutive models, collision handling, and fracture is an important direction, and we will explicitly highlight it as future work rather than implying that the current system already covers these phenomena.
>
> > **Reviewer Tp4o concerns  scalability to cluttered, real-world scenes where SuperGaussian clustering and constraint graphs may become unreliable..**
>
> Thanks. We agree that pipeline in current version focus on **single, relatively isolated objects**, primarily for clarity of visualization. However, both our **dataset choice** and **algorithmic design** are intended to support more complex scenarios:
>
> - Beyond NeRF-style synthetic scenes, we also experiment on **ShapeSplatsV1**, which contains on the order of **65K Gaussian-splat objects** across **87 categories**. This dataset provides substantially more geometric and structural diversity than a handful of toy examples.
> - Our SuperGaussian clustering and constraint-graph construction are explicitly **local and sparse**: each Gaussian or cluster only connects to nearby neighbors under a combined spatial–appearance metric, and edges are thresholded by per-node statistics (`cf. Sec. 3.3.2, Sec. 3.4.2`). This design yields *near-linear* complexity rather than quadratic and avoids noisy long-range links, which helps scalability as the scene becomes more complex.
>
> In the rebuttal and camera-ready version, we will:
>
> - Add **additional qualitative examples** from ShapeSplatsV1 showing objects with multiple interacting parts and more complex shapes.
> - Include these examples as **video demos** on the anonymous project page (https://anonymous.4open.science/w/animating-page-0F98/) to better illustrate robustness beyond very simple scenes.
>
> We agree that fully cluttered, multi-object real-world scenes with heavy occlusion and contact are more challenging and form an important direction for future work. We will explicitly clarify that our current focus is *object-level 3D cinemagraphs with clean geometry*, a setting commonly adopted in recent 3DGS-based works, and leave fully cluttered multi-object scenes as a future extension.
>
> > **Reviewer Tp4o concerns about high-level user-control**
>
> Good point !  Our current notion of “user control” is parametric and physically grounded, rather than fully semantic:
>
> The user specifies a 3D force field (e.g., wind) by tuning parameters such as direction, magnitude, spatial falloff, and spatial region (e.g., above a height threshold, within a 3D box).
>
> This field is evaluated at SuperGaussian centers, so different spatial regions of the same object can exhibit different motion intensities (e.g., upper branches vs. trunk), even though the underlying force model is global and parametric. We conduct a toy example of force field generation in (https://anonymous.4open.science/w/animating-page-0F98/).
>
> We will explicitly reference these capabilities and examples in the revised manuscript.

---

> ### Comment · Reviewer_Tp4o · 2025-11-26
>
> I appreciate the authors for providing the complete rebuttal, and supply video demos as I asked.
> Somehow, I could not view many of the videos on the anonymous webpage, and when I refresh I get "You can only make 350 requests every 15min. Please try again later.".
> From the videos I saw, it seems when forces are applied, there is always an overall "stretch" on the entire scene? Is there a cause of this?
>
> (I understand this is a very difficult task and would be happy to raise my score to a 6.)

---

> > ### Author Response · Authors · 2025-11-26
> > **Response to R#Tp4o**
> >
> > We sincerely thank the reviewer for carefully going through our rebuttal and additional demos, and we greatly appreciate your willingness to raise the score.
> >
> > **On the video access issue.**
> >
> > We apologize for the inconvenience caused by the anonymous hosting service and its strict rate limitation (*“You can only make 350 requests every 15min”*). To avoid any further throttling or loading issues, we have prepared an anonymous downloadable package at https://limewire.com/d/x64lu#QqyzE8unPl which is valid throughout the rebuttal period. After downloading, all demos can be viewed locally by simply opening the *index.html* file in a browser; this page is fully static and does not collect or reveal any identity information.
> >
> > For ease of navigation, the video set is organized into four parts:
> >
> > **(1) Basic demos on simple scenes**, illustrating the core behavior of our method under different force configurations.
> >
> > **(2) Comparisons with existing methods** on two representative scenes.
> >
> > **(3) User-interactive force-field construction**, showing how user-specified parameters define a 3D force field (visualized on three orthogonal planes) and how this field is applied to the scene.
> >
> > **(4) Two distinct force fields and their mixture**, demonstrating how different and combined force fields lead to different deformation behaviors.
> >
> > **On the apparent “overall stretch” of the scene.**
> >
> > The slight global “stretching” you observed when forces are applied is a side effect of how our deformation model is regularized: We model forces as a continuous field defined over the full Gaussian set, coupled with a soft elastic regularizer that encourages neighboring Gaussians to move coherently for stability. Static/background regions are therefore not enforced as perfectly rigid by a hard constraint; instead, they are assigned large but finite stiffness. Under strong or spatially broad forces, this can induce a small low-frequency drift over the entire configuration, which visually appears as a mild global stretch. Importantly, this effect is bounded and controllable: by increasing the stiffness/anchor weights of background Gaussians, we can substantially suppress this drift while keeping the motion stable. In the updated demos, we have tuned these parameters so that the foreground remains deformable under user forces, whereas the background and static structures stay nearly rigid.
> >
> > We hope this clarifies the cause of the artifact you noticed, and we again thank you for the constructive feedback and the positive overall assessment of our work.

---

### Author Response · Authors · 2025-12-01
**A summary note at the end of author–reviewer discussion period**

Dear PCs, SACs, ACs, and Reviewers,

Thank you for overseeing the reviewing process. The author-reviewer discussions make the threads go too long for you to quickly grasp the state of our rebuttal. Therefore, we drop this message to summarize our paper, the reviews, and our rebuttal.

**Summary of our paper**

Our work proposes a physics-driven framework for *3D cinemagraphs built on 3D Gaussian Splatting*. We endow each Gaussian with physical attributes (e.g., mass) and evolve the scene under *Newton’s second law* driven by external and internal forces. This enables dynamic, loopable 3D animations from reconstructed scenes while maintaining *physical plausibility, structural stability, and real-time 3DGS rendering*.

Concretely, we:
- Formulate a *mass-augmented 3D Gaussian representation* and a differentiable dynamics model integrating multiple external forces with elastic regularization.
- Introduce *user-controllable 3D force fields*, parameterized and applied to the Gaussian set for intuitive, localized motion editing.
- Design a training/regularization strategy that preserves global structure and appearance while allowing plausible non-rigid deformation, yielding temporally coherent 3D cinemagraphs.

Experiments on multiple scenes show that our method produces *more realistic and physically consistent motion* than prior 3DGS-based and MPM-based methods, with practical training time (≈10 minutes on a single RTX 4090) and real-time rendering.

**Summary of reviews and rebuttal**

All four reviewers regard our work as `highly novel`. They emphasize that 3D cinemagraphs are “an interesting and visually appealing research topic,” and that **assigning mass attributes to Gaussians and modeling motion via Newton’s second law provides a new, physically grounded perspective** beyond previous MPM-based approaches. All reviewer explicitly describe the method as physically well-motivated and controllable.

The main concerns focus on presentation and clarity rather than the core idea:

- **Video results and clarity.**
  Reviewers requested clearer and richer video demonstrations. In the rebuttal, we have provided an anonymous webpage (https://anonymous.4open.science/w/animating-page-0F98/) and a downloadable, self-contained video package (viewable locally via a static `index.html` without any identity information <https://limewire.com/d/x64lu#QqyzE8unPl>). The demos include simple-scene examples, comparisons with baselines, user-interactive 3D force-field construction, and different/combined force fields, making the motion behavior and user control easier to assess.

- **Experimental details and quantitative evaluation.**
  Some reviewers confused about existing qualitative results and asked for more results. We welcome the suggestions and have added *more comparison scenes, and additional ablations* on the anonymous webpage.

- **“Overall stretch” / global drift.**
  One reviewer observed an apparent global “stretch” when forces are applied. We explained that this arises from our **soft elastic regularization**: static regions are given large but finite stiffness rather than hard rigidity, so strong/broad forces can induce a small low-frequency drift. We showed that this effect is **bounded and controllable** via stronger background anchoring, and provided updated demos where foreground deforms while background remains nearly rigid, noting this trade-off and future rigid–deformable decomposition.

- **User control and computational cost.**
  We clarified how user-specified parameters define spatially varying 3D force fields that can be localized to specific regions, and reported that training per reconstructed scene takes ≈10 minutes on an RTX 4090, with real-time inference thanks to standard 3DGS rasterization.

After the discussion, at least one reviewer explicitly indicated a willingness to **raise the overall score**. For the remaining points, we believe the additional experiments, videos, and explanations have substantially addressed the reviewers’ concerns. Although the OpenReview incident prevented further replies from some reviewers, we warmly welcome any additional questions and would be happy to clarify further to help the committee and reviewers reach a clear consensus.

We sincerely thank the SACs, ACs, and Reviewers again for their time and constructive feedback, and we hope this concise summary is helpful for your deliberation.

Regards,
Authors of Paper-6483

---

### Meta-Review · Area_Chair_nD8U · 2026-01-07

**Summary:**

The reviews agree the idea is interesting, but there is significant concern that the evidence is not strong enough for ICLR: (i) weak/limited experimental validation and generalizability, including requests for more scenes and stronger evaluation, (ii) missing or insufficient comparisons, including calls for more recent physics-integrated baselines (e.g., OmniPhysGS), and (iii) lack of video demonstrations in the original submission, which reviewers felt was essential for judging motion realism/temporal coherence. After reading the paper and watching the rebuttal materials, the AC still finds the resulting animations not compelling enough overall, and doesn’t think the paper meets the ICLR bar.

**Reviewer Concerns:**

Addressed by rebuttal:
(1) The “no video” concern: authors provided a project webpage with video comparisons/demos
(2) Some clarification items (e.g., what was/was not a formal user study) were explicitly clarified

Still outstanding:
(1) The core concern of insufficient experimental validation (limited scenes, limited breadth) remains a major weakness
(2) Requests for stronger / more recent baselines were not fully resolved in the rebuttal (e.g., OmniPhysGS was recommended)
(3) Even with videos added, the qualitative results did not convincingly demonstrate a clear, consistent advantage at the level expected for acceptance.

**Reviewer Scores:**

TbCs (2): likely stays at 2 given the primary complaint is weak/insufficient evaluation and missing comparisons

b62R (4): likely stays around 4; videos address a key presentation issue, but the review also cites unclear results and insufficient evaluation breadth

Tp4o (4): likely increases to 6, since the reviewer explicitly indicated willingness to raise the score after seeing the provided video demos (despite access/friction and remaining artifacts)

rURL (6): likely stays at 6; the review was already slightly positive but still wanted newer baselines and better video presentation

---

### Decision · Program_Chairs · 2026-01-26

Reject